# Electrocardiograph Identification Using Hybrid Quantization Sparse Matrix and Multi-Dimensional Approaches

**DOI:** 10.3390/s18124138

**Published:** 2018-11-26

**Authors:** Kuo-Kun Tseng, Jiao Lo, Chih-Cheng Chen, Shu-Yi Tu, Cheng-Fu Yang

**Affiliations:** 1School of Computer Science and Technology, Harbin Institute of Technology, Shenzhen 518055, China; kktseng@hit.edu.cn (K.-K.T.); jiaoluo126@126.com (J.L.); 2School of Information Engineering, Jimei University, Xiamen 361021, China; 3Mathematics Department, University of Michigan, Flint, Ann Arbor, MI 48502, USA; sytu@umflint.edu; 4Department of Chemical and Materials Eng., N.U.K., Kaohsiung 81148, Taiwan

**Keywords:** quantization sparse matrix, ECG, blood oxygen, identification

## Abstract

Electrocardiograph (ECG) technology is vital for biometric security, and blood oxygen is essential for human survival. In this study, ECG signals and blood oxygen levels are combined to increase the accuracy and efficiency of human identification and verification. The proposed scheme maps the combined biometric information to a matrix and quantifies it as a sparse matrix for reorganizational purposes. Experimental results confirm a much better identification rate than in other ECG-related identification studies. The literature shows no research in human identification using the quantization sparse matrix method with ECG and blood oxygen data combined. We propose a multi-dimensional approach that can improve the accuracy and reduce the complexity of the recognition algorithm.

## 1. Introduction

Human identification has become an important issue in many areas, with people more aware than ever before of the need to protect data and prevent information from leakage, theft, or unauthorized modification. Many widespread electronic techniques and approaches, such as fingerprinting and iris scanning, are now used extensively in real-world applications. In medicine, electrocardiograms (ECGs) and blood oxygen levels are widely accepted methods for monitoring heart disease. Recently, ECGs have also been employed extensively in the field of security studies, particularly to verify human identity for access control systems [1,2,3,4].

The ECG method, a technique for measuring and recording different electrical potentials in the heart, is a powerful tool, but most recognition algorithms utilize ECG signals without other biometric characteristics [5,6], ignoring potentially useful additional information. A comprehensive survey of the literature revealed no studies combining ECGs and blood oxygen levels for identity verification. In this research, we present a new identification system that uses an algorithm combining ECG signals and blood oxygen levels, showing that they work together well to verify human identity. Further, the requisite data are easily collected from a fingertip through a wearable device. Identification using these biometrics is based on multidimensional analysis in which both data are mapped into a two-dimensional coordinate system and the intervals between points are quantized by counting their values. All entries in the quantized sparse matrix are used to find the correlation coefficient, an index for determining the success or failure of the identification. We call the proposed algorithm the Quantization Sparse Matrix Identification (QSMI) algorithm, since it relies heavily on ideas of quantization and the application of sparse matrices.

The previous method can be considered a two-dimensional algorithm. We can increase the number of dimensions to improve recognition accuracy. The Multi-Dimensional Identification (MDI) algorithm greatly increases the computational complexity, so we propose an improved algorithm combined with the Dynamic Time Warping (DTW) algorithm, which can reduce the computational complexity by effectively increasing the recognition rate, making multi-dimensional ECG recognition more feasible. The rest of the paper is organized as follows. Past research in multi-dimensional algorithms is reviewed in Section 2; the proposed algorithm is introduced in Section 3; experimental results and comparisons with other models are presented in Section 4; lastly, a summary of our comparisons and a discussion are provided in Section 5.

## 2. Related Work

No two individuals have exactly the same biological characteristics, so each person’s ECG signals are unique [7]. ECGs are widely used in diagnostics [8,9], compression [10], watermarking [11,12], and verification [3,4,5,6,8]. Numerous studies have sought to establish human identify using one-lead ECG signals [13,14], but not much attention has been paid to employing ECGs with other forms of biometric information. Yet identification requires strong reliability and high accuracy, so the use of more biometric information typically results in a better identification rate.

In 2001, Masaki and Akihiko [15] improved an identification engine using multiple discrimination analysis. Edward proposed blending standard 12-lead ECG signals for identification purposes [16]. Multi-dimensional analysis [17] is an informational method that examines many different relations between data, each of which represents a dimension. Commonly used in statistics, economics, and other related fields [18], this process groups data into two categories: dimension and measurement. Michael [19] analyzed the effects of industry, region, and time on new business survival rates using a multi-dimensional approach. The research in [20] focused on multiple dimensional analysis of stream data and related this to time series analysis.

Blood oxygen level measures how much oxygen red blood cells contain [21]. A blood oxygen test is usually performed to examine breathing disorders or other conditions that may cause decreased oxygen levels in a human body. Jackson III [22] in 1998 invented a device to monitor infants’ blood oxygen levels and to prevent sudden infant death syndrome. Blood oxygen levels have also been used in disease diagnosis [23] and prognosis [24] studies. Past research shows that most human recognition schemes rely only on feature extraction from ECGs without employing other forms of biometrics. Vanni used ECG signals and blood pressure to detect acute hypertensive episodes and mean arterial pressure dropping regimes [25]. Syeda and Varun [26] investigated sleep apnea with both ECG signals and blood oxygen levels. However, no human identification research has employed both types of data. In this study, we link ECG and blood oxygen to propose an algorithm for verifying human identity.

## 3. Proposed Model

### 3.1. Basic Model

The study aims to enhance human identification accuracy based upon two types of biometric information fed into the proposed QSMI algorithm. In our scheme, the x- and y-axes correspond to the data we are relating, namely, ECG signal and blood oxygen level, respectively. Each ordered pair in the two-dimensional coordinate system uniquely represents one individual’s characteristics at a specific time. The flowchart of the QSMI algorithm is show in Figure 1. 

Consider two sets of data, ECG signal and blood oxygen level, collected from an individual i. We write
(1)$$E_{i}=\left\{e^{i}\left(t_{1}\right),\;e^{i}\left(t_{2}\right),\cdots ,e^{i}\left(t_{k}\right)\right\}$$
$$B_{i}=\left\{b^{i}\left(t_{1}\right),\;b^{i}\left(t_{2}\right),\cdots ,b^{i}\left(t_{k}\right)\right\}$$
where $e^{i}\left(t_{k}\right),b^{i}\left(t_{k}\right),\;k=1,\cdots ,n$ denotes the ECG and blood oxygen sample values obtained from the *i*th individual at a specific time $t_{k}$. Describe the dataset using a two-dimensional coordinate representation, namely, $M_{i}\left[e^{i}\left(t_{k}\right),\;b^{i}\left(t_{k}\right)\right]$. Without loss of generality, $M_{i}$ is initially set to be 0. According to the mapping rule, we have

(2)$$Counting\;():\;M_{i}\left[e\left(t_{k}\right),\;b\left(t_{k}\right)\right]=M_{i}\left[e\left(t_{k}\right),\;b\left(t_{k}\right)\right]+1,\;\forall \exists \left(e\left(t_{k}\right),\;b\left(t_{k}\right)\right)$$

$$\mathrm{Set}\;v_{k}^{i}=M_{i\;}\left[e\left(t_{k}\right),\;b\left(t_{k}\right)\right]iff\;Counting()=end$$

Let $v_{k}^{i}$ be the value at the element [$e^{i}\left(t_{k}\right),\;b^{i}\left(t_{k}\right)]$ when counting processing is end; this is a counting process to map two time series into a matrix.

After the mapping processing is finished, the combination of ECG signal and blood oxygen level can be expressed in the format of a diagonal matrix M of size n×n. We now reduce the size of *M* using the quantization method. Define a mask matrix $M^{*}$ of size m×m, where *m* is the dimension of the mask matrix for quantization; $M^{*}$ is initially set as a zero matrix. m×m size is also an reduction rating. First, $k_{x}$ is from 1 to n and $k_{y}$ from 1 to n of original matrix M, we use matrix $M^{*}$ to calculate the sum of window m×m. The result store as the element of new reduction matrix RM[]. In this way, we follow the rule:(3)$$RM\left[\frac{k_{x}+m-1}{n},\frac{k_{y}+m-1}{n}\right]=$$

$$\{\begin{array}{c} \begin{array}{c} 0,\;M\left[k_{x}:k_{x}+m-1,k_{y}:k_{y}+m-1\right]+M^{*}\leq a\\ 1,\;b<M\left[k_{x}:k_{x}+m-1,k_{y}:k_{y}+m-1\right]+M^{*}\leq c \end{array}\\ \begin{array}{c} 2,c<M\left[k_{x}:k_{x}+m-1,k_{y}:k_{y}+m-1\right]+M^{*}\leq d\\ 3,\;d<M\left[k_{x}:k_{x}+m-1,k_{y}:k_{y}+m-1\right]+M^{*\;} \end{array} \end{array}$$

The variables *a*, *b*, *c*, and *d* are the quantization levels we defined, as shown in Figure 2. In addition, $M\left[k_{x}:k_{x}+m-1,k_{y}:k_{y}+m-1\right]$ means the field of the horizontal ordinate X is from $k_{x}$ to $k_{x}+m-1$. The same is true for the vertical ordinate Y.

After reduction, RM[] is a matrix with high sparsity since most of its entries are zero. Sparse data are easily compressed and use less storage space; it is beneficial and often necessary to store only the non-zero entries. There are numerous representations for sparse storage; in this study, we use the simplest and most flexible coordinate format (COO), which lists the row, column, and value as a 3-tuple. For example, information in the reduced matrix RM[] in Table 1 can be stored as {(2, 2, 1), (1, 3, 2), (2, 3, 2), (3, 3, 2), (1, 4, 2), (2, 4, 2), (1, 5, 1), (2, 5, 1)} as the elements of sparse matrix SM[].

After biometric data are collected from an individual, the information is transformed immediately into a sparse matrix and is stored using COO without any pre-processing. A sparse matrix SM[], stores the features of the data, and its entries are inputs to the correlation coefficient classifier. The correlation coefficient between two SM[] will be calculated for future training and identification purposes, serving as an index to train and test the accuracy of human verification with an appropriately selected threshold. Correlation coefficients are widely applied in the sciences to determine how strongly units from the same group resemble each other; Mathworks documentation (http://www.mathworks.com/help/matlab/ref/corrcoef.html) describes this in detail.

### 3.2. Advanced Model

The advanced model we propose is a combination of the reduced sparse matrix of MDI and DTW methods. This method of mapping multi-lead signal data into multi-dimensional space and extracting sparse features can achieve better matching results. With equipment used to measure ECG signals, 12-lead or 5-lead are the most commonly used methods, which are more suitable for MDI. The main difference from basic model QSMI, MDI reduces input signal level at first; it can greatly reduce the data complexity of high dimensional matrix.

Then, DTW can compute the similarity between two time series, especially for different lengths and different rhythms of time series. DTW automatically warps distorted time series to make the two sequences as consistent as possible and obtain the greatest possible similarity.

Although multi-dimensional data can represent more information, there will be redundant information, so we need to reduce the dimensionality of the multi-dimensional data and enhance the extraction of feature information. Our advanced model relies on an algorithm that fuses multi-dimensional sparse matrix and DTW. For a large range of ECG data, in order to better aggregate the expression of the data features we first reduce the data to a certain range, then map them to a multi-dimensional space. There are two stages in our advanced model, as follows.

In the MDI stage, it wants to map and reduce multiple lead signals into a reduced sparse matrix. According to the multi-dimensional space algorithm mentioned above, we assume that $s_{i}^{j}\left(t_{k}\right)$ is the ith individual for sampling of the jth lead at time *t* of the sample point value. For reduced matrix processing, we first map multiple time series points $s_{i}^{j}\left(t_{k}\right)$ for each lead *j* at time *t*, and we map the signal value to the *j*-dimensional coordinate point $p_{i}\left(t_{k}\right)=\left(d_{i}^{1}\left(t_{k}\right),\;d_{i}^{2}\left(t_{k}\right)\ldots ,d_{i}^{j}\left(t_{k}\right)\right)$ according to the following formula:(4)$$d_{i}^{j}\left(t_{k}\right)=\{\begin{array}{c} \frac{s_{i}^{j}\left(t_{k}\right)}{R}+∆\theta ,\;else\\ 1\;,\frac{s_{i}^{j}\left(t_{k}\right)}{R}+∆\theta <1\\ u,\;\frac{s_{i}^{j}\left(t_{k}\right)}{R}+∆\theta >u \end{array}$$
where point $p_{i}\left(t_{k}\right)$ is constructed by *j* dimensional reduction value $d_{i}^{j}\left(t_{k}\right)$, *R* is a reduction level we define, and ∆θ is an increment to prevent negative point values. The result of the second and third lines is to marginalize the result and prevent the map from overflowing, where *u* is the maximum point value of the space we map.

In this way, a multi-dimensional coordinate space is applied, then the multi-lead signal is mapped to the space according to time $t_{k}$; next, reduced sample point processing is performed, and finally the spatial correlation coefficient is calculated to complete the recognition judgment. In the next part, we will show the experimental evaluation results for this algorithm. Since high-dimensional space occupies a large amount of memory, storage is inconvenient. We therefore need to build a reduced sparse matrix to simplify the calculation and storage of the sample coordinates of the multi-dimensional space.

After a reduced sparse matrix is constructed, we upgrade the idea of a two-dimensional sparse matrix to a multi-dimensional coordinate space. For example, if the value of the *j* dimensional coordinate point $\left(d_{i}^{1}\left(t_{k}\right),\;d_{i}^{2}\left(t_{k}\right)\ldots ,d_{i}^{j}\left(t_{k}\right)\right)$ with value *v*, then the sparse element we store is $\left(\left(d_{i}^{1}\left(t_{k}\right),\;d_{i}^{2}\left(t_{k}\right)\ldots ,d_{i}^{j}\left(t_{k}\right)\right),v\right)$ The subsequent correlation calculation no longer uses the correlation coefficient between the matrices, because the correlation coefficient of the high-dimensional coordinate space is calculated, the time and space complexity are too high to be practically applied, and we will extract a series element of high-dimensional sparse matrix. The sparse matrix is used as the input to the DTW for calculating the correlation between the two series. The algorithm for integrating our multi-dimensional feature with DTW is described in detail below.

As mentioned, a training or test sample for each individual *i* is ultimately represented as a representation of a series of sparse elements. Assume that the *s*th sample template of the *i*th individual is stored as $X_{i}^{s}$, then the algorithm.
(5)$$X_{i}^{s}=\left(p_{i}^{s}\left(t_{1}\right),p_{i}^{s}\left(t_{2}\right),\ldots ,p_{i}^{s}\left(t_{k}\right)\right)$$
among them, the distance can be used as a metric:(6)$$D\;\left[X_{i}^{s},\;X_{i^{\prime }}^{s^{\prime }}\right]$$

Here is two high dimensional sequences, where *s* and $s^{\prime }$ mean any two sample templates of two *i* and $i^{\prime }$ indiviuals. In the DTW stage, the two test templates are represented as $X_{i}^{s}$ and $X_{i^{\prime }}^{s^{\prime }}$. To calculate the similarity between them, the smaller the distance, the higher the similarity between $X_{i}^{s}$ and $X_{i^{\prime }}^{s^{\prime }}$. Let k and $k^{\prime }$ be the sequence lengths of $X_{i}^{s}$ and $X_{i^{\prime }}^{s^{\prime }}$, respectively, so $D[X_{i}^{s},\;X_{i^{\prime }}^{s^{\prime }}$] represents the distance between these two sequences. If:
(1)$D\;\left[X_{i}^{s},\;X_{i^{\prime }}^{s^{\prime }}\right]$; $k=k^{\prime }$, directly calculate *D*$\left[X_{i}^{s},\;X_{i^{\prime }}^{s^{\prime }}\right]$ by Euclidean distance;(2)$k\neq k^{\prime }$, use dynamic programming to calculate the distance between them.


DTW is described in detail in [27]. We only need to apply this idea to the minimum distance to find two different sequences, which will not be described here. Multi-dimensional DTW is calculated in a similar way to DTW for single-dimensional time series, except that we redefine $D\left[X_{i}^{s},\;X_{i^{\prime }}^{s^{\prime }}\right]$ as the cumulative squared Euclidean distances of multiple data points instead of the single data point used in the more familiar one-dimensional case. For details, please refer to [28].

## 4. Experiments and Evaluations

A reliable biometric identification system is capable of handling five basic elements: data collection, transmission, signal processing, storage, and decision making. Past research has shown that even with ECG or other biometric-based verification algorithms, no practical solutions have been found to cut the cost of implementation. To overcome this obstacle, we implement a portable device, as shown in Figure 3, so the identification process can be done efficiently and less expensively. This board uses a TI AFE4900 chip, and the AFE4900 device is an analog front-end (AFE) for synchronized ECG and photoplethysmogram (PPG) signal acquisition. The device can also be used for optical bio-sensing applications, such as monitoring heart rate and measuring the saturation level of peripheral capillary oxygen (SpO_2_).

ECG signals and blood oxygen level data can be collected by placing this small sensor on a thin area of an individual’s body, usually a fingertip; signals are then transmitted to the user’s mobile phone via Bluetooth and stored directly in its memory card. This equipment provides a low-cost, effortless way to gather and handle biometric information necessary for human recognition. Given the popularity of wearable monitoring devices [29,30], we believe that our equipment has the potential for wide application.

### 4.1. Experiment for QSMI Algorithm

For our verification experiment, data were collected from 18 individuals. This raw dataset contains a sequence of point values at certain times. For each individual, two sets of data were collected, one for training purposes and the other for testing the proposed algorithm.

We have two training set for ECG and blood oxygen signals data; the rest of the set is for the test samples data. In the training stage, each set of one individual should be a template matrix compared to every other set, including set of different individuals. In the training stage, we use the training data set of all 18 individuals to evaluate the correlation coefficients. To verify the efficiency of this algorithm, we train the sparse matrix containing a segment of sample data points and get thresholds for each individual.

Numerous metrics have been used to assess the performance of a biometric factor. The most common performance metrics—false acceptance (FA) and false rejection (FR) rates—are employed to evaluate the QSMI algorithm. The FA rate, which is the probability that a biometric system will accept an incorrect input as a positive match, is obtained by testing known biometric templates against a large data collection. By contrast, the FR rate represents the probability that a biometric system will incorrectly reject an input as a negative match. FA and FR rates usually show an inverse relationship with one another. We define the accuracy rate, Acc, as

(7)$$Acc=1-\frac{FA+FR}{2}$$

The purpose of a threshold is to determine how close to a template the input data must be for it to be considered a match. The threshold for $\frac{\mathrm{FA}+\mathrm{FR}}{2}$ will be obtained from the training set; this procedure will be discussed in detail later. Figure 4 illustrates the relationship between $\frac{\mathrm{FA}+\mathrm{FR}}{2}$ and *δ* for the classification procedure. It shows that $\frac{\mathrm{FA}+\mathrm{FR}}{2}$ lies between 0.0686 and 0.2059 in template $T_{1}$, and between 0.1830 and 0.2386 in template $T_{2}$. A better Acc rate can be attained using an iterative approach to reach a suitable threshold δ.

As we can see from (5), a smaller $\frac{\mathrm{FA}+\mathrm{FR}}{2}$ value yields a higher Acc. Figure 4 shows that template $T_{1}$ has a better accuracy rate, so only $T_{1}$ will be used for discussion. Acc is maximized as $\frac{\mathrm{FA}+\mathrm{FR}}{2}$ reaches its minimum, when δ=0.019, so the appropriate threshold δ for multiple subject classification is set at 0.019. Figure 5 shows detailed rate information for template $T_{1}$ as δ increases from 0.001 to 0.03, clearly showing that $\frac{\mathrm{FA}+\mathrm{FR}}{2}$ reaches its minimum when δ=0.019.

The FA rates of all 18 individuals in $T_{1}$ when δ=0.019 are shown in Figure 6. Since FR equals 0 for all individuals, it is not displayed.

Since the sample size of our ECG and blood oxygen data is small, it is highly possible to have the data map to the same point. To take advantage of this feature point, mapping rule (2) is employed to define the quantization interval. To achieve optimal results, various quantization intervals are assessed, and the experimental results of the improved algorithms are shown in Figure 7. Since all FR rates are 0 for all quantization intervals, only FA and $\frac{\mathrm{FA}+\mathrm{FR}}{2}$ are presented.

As can be seen in Figure 7, the $\frac{\mathrm{FA}+\mathrm{FR}}{2}$ rate is 0.1193 when no quantization is carried out and reaches its minimum, 0.0686, when a three-level (0, 1, 2) quantization procedure is performed for data points reduction. More specifically, implementing the proposed QSMI algorithm raises the Acc rate for identification from 0.8807 to 0.9314, an approximately 5% enhancement.

### 4.2. Experiment for MDI Algorithm

First, to prove the correctness of our multi-dimensional algorithm, we slowly and incrementally use the same evaluation method for the PTB data (see, for example, the PTB database at www.physionet.org/physiobank/database/ptbdb/), from the double-lead ECG sensor, obtaining the results shown in Figure 8.

It can be seen from Figure 8 that when the lead dimension is increased, $\frac{\mathrm{FA}+\mathrm{FR}}{2}$ is smaller, that is, the accuracy is higher. When the dimension reaches 5, $\frac{\mathrm{FA}+\mathrm{FR}}{2}$ is 0.013. At this point, the error rejection rate is 0, and Acc is up to 98.7%, proving that our multi-dimensional feature algorithm is feasible when mapping multi-dimensional feature data to space.

In the description of the MDI algorithm, we introduce the reduction remapping first, and the variation in the reduced parameter variable ∆*θ* affects the recognition rate. Figure 9 shows the change in ∆*θ* with the $\frac{\mathrm{FA}+\mathrm{FR}}{2}$ curve.

From Figure 9 we see that the FA/FR rate of the recognition result changes as ∆*θ* changes. When ∆*θ* = 20, $\frac{\mathrm{FA}+\mathrm{FR}}{2}$ reaches 0.0133, which is the best result for 5D, as shown in Figure 8. Of course, the 2D to 4D ∆*θ* in Figure 8 are the best values given under our experiment, and the chart description is no longer listed here.

According to our previous charts, when the number of leads is 5—that is, the computational space is 5 dimensions—the experiment achieves good results but takes up a lot of memory. Hence, we need to find a new algorithm to optimize the data. To do so, we introduce the method of using DTW and sparse combination to solve the similarity calculation problem of different sequence lengths. Figure 10 presents the experimental results using DTW and sparse combination in a 5-dimensional space.

In Figure 10, Num_n_Red_m on the abscissa indicates that the size of the window space of size *m* is reduced when the space size is *n*. It can be concluded from the figure that when the 5-lead ECG signal is mapped to a 5-dimensional space of 32 and the window is reduced by 2, the recognition result $\frac{\mathrm{FA}+\mathrm{FR}}{2}$ is the smallest, 0.0115, slightly higher than the multi-dimensional feature recognition (baseline) does not combine with DTW sparse algorithm, which will generate experimental result of 0.0133. We evaluate the spatial and temporal efficiencies of several of the above-mentioned algorithms using a number of experimental data and averaging the methods to evaluate them fairly. Table 2 compares the average memory space occupied by the DTW and the sparse algorithm, and the average memory space of the baseline algorithm.

When MDI is combined with the DTW sparse optimization algorithm, because the number of sparse eigenvalues of each template sample is inconsistent, the memory space they occupy is also different. Table 2 lists all the samples with their average, minimum, and maximum memory space, but the occupied memory space of each sample is no longer listed. As can be seen, when MDI is combined with the DTW sparse algorithm, the average memory space for all the samples is much smaller than for the baseline algorithm. In actual applications, the maximum memory space required is an obvious factor. The original DTW uses the most memory. In comparison, the memory required for the new method is much smaller than for DTW and much smaller than for the baseline, so an identification system with this algorithm could be used in an embedded system or on a small chip.

However, when designing the algorithm, we should consider not only the spatial complexity but also the time complexity, so we evaluate the above algorithm in terms of time efficiency. Table 3 compares the time required to complete the experiment when using DTW in combination with the sparse method for identification.

Using Figure 10 and Table 3, we can conclude that when the DTW is combined with the sparse algorithm, the time efficiency decreases as the recognition accuracy increases. This is because the dynamic programming algorithm is used when the final reduced feature value is increased such that the time efficiency rises to the power of $O\left(n^{2}\right)$. In the baseline algorithm, dynamic programming is not used, and only the correlation coefficient between the two samples is calculated, so the time complexity is not as high as with the MDI algorithm. These experimental results show that the MDI algorithm and the DTW combined with the sparse identification multi-dimensional feature algorithm have advantages and disadvantages in terms of time and space efficiency, but the MDI algorithm increases the time by only 1 or 2 seconds in comparison with the baseline algorithm, so it is still very feasible for practical applications.

## 5. Comparison

### 5.1. Comparison with One-Dimensional Models

To date, not much attention has been paid to investigating human identity verification using a combination of ECG signal and blood oxygen level. In this section, three designs involving only ECG signals are introduced. We then compare our scheme with these one-dimensional algorithms. To achieve a fair assessment, all data are selected from the same database—the biometric collection gathered from the wearable device introduced in Section 4.

(a) Reduced binary pattern (RBP)

The reduced binary pattern algorithm employs the frequency and rank order statistics of the input ECG signals [31]. Express any collected ECG signal from an individual in the form $\left\{x_{1},x_{2},x_{3},\ldots ,x_{n}\right\}$, where each real-value $x_{i}$ denote the ith value from the input data. Compare any two consecutive values and categorize the data into two cases: decrease or increase in $x_{i}$. A preliminary reduced function then maps these two cases to 0 or 1, respectively, according to the following two-state rule as $y_{i}=\{\begin{array}{c} 1\;,\;x_{i+1}\geq x_{i,}\\ 0,\;otherwise. \end{array}$

This step converts one segment of the ECG signal of length n to a binary sequence $Y=\left\{y_{1},\;y_{2},\ldots ,y_{n-1}\right\}$ of length n−1. Group every m bits in Y into a reduced binary sequence and collect all m-bits to form a reduced binary pattern $\left\{b_{1},\;b_{2},\ldots ,b_{n-m}\right\}$, where $b_{k}=\left\{y_{k},\;y_{k+1},\;\ldots ,y_{k+m-1}\right\}$.

Let $d_{k}$ be the decimal expansion of $b_{k},\;k=1,\;2,\;\cdots ,\;n-m$. It is obvious that the values of $d_{k}$ can change from 0 to $2^{m}-1$. Count the occurrences of each $d_{k}$, sort them in order of descending frequency, and find the corresponding probability. Ranking can be omitted if the tested ECG segments have the same sampling duration.

Consider two segments of ECG data *S*_1_ and *S*_2_, which may belong to two distinct individuals. To determine how closely they are related, the measurement of similarity between $S_{1}$ and $S_{2}$ is defined as
(8)$$D_{m}\left(S_{1},S_{2}\right)=\frac{1}{2^{m}-1}\cdot \frac{\sum_{d_{k}=0}^{\left(2^{m}-1\right)}\;\left|R_{1}\left(d_{k}\right)-R_{2}\left(d_{k}\right)\right|\;p_{1}\left(d_{k}\right)\;p_{2}\left(d_{k}\right)}{\sum_{d_{k}=0}^{\left(2^{m}-1\right)}\;p_{1}\left(d_{k}\right)p_{2}\left(d_{k}\right)}$$
where $p_{i}\left(d_{k}\right)$ and $R_{i}\left(d_{k}\right)$ represent the relative frequency and rank of $w_{k}$ in the sequence $S_{i},\;i=1,\;2$. The absolute difference between two ranks is multiplied by the normalized probabilities as a weighted sum; the factor $\frac{1}{2^{m}-1}$ ensures all values of $D_{m}$ lie within the scope of (0, 1).

(b) Waveform algorithm.

Waveform algorithms [32] use characteristic features obtained directly from the ECG waveform to verify human identity. After an ECG waveform is measured, it passes through high- and low-pass filters for pre-processing. Various techniques are then carried out to extract the characteristic points inside the waveform where ECG complexes P, Q, R, S and T are located. Based on the extracted values of the characteristic points, some identification features are composed of relative representations. The extraction of characteristic points and the composed features are shown in Figure 11. In a waveform-based study [33], a total of 19 features were extracted from four classes: amplitude (PQ, RQ, TQ, RT, PS, RP, TS, RS, PT, QS), duration (QS, PR, QR, ST, QT), slope (RS, ST, QR), and area (area of the QRS triangle). These features form a feature-vector S.

Based on the feature-vectors obtained from the individuals, we use the measurement of similarity formula (6) to evaluate the difference between two subjects. Closeness between two feature-vectors $S_{1}$ and $S_{2}$ is denoted as the distance $d\left(S_{1},S_{2}\right)$. We then use (5) to calculate the FA, FR, and $\frac{\mathrm{FA}+\mathrm{FR}}{2}$ rates of all obtained distances.

(c) Wavelet transform algorithm

A wavelet transformation is the representation of a function by wavelets, which are scaled copies of finite-length or fast-decaying oscillation waveforms. The wavelet-based algorithm [34] includes the following procedures: each R-R cardiac cycle is obtained through R-R detection; an interpolation is performed on the R-R interval so each R-R cardiac cycle holds 284 data points; every R-R cycle is cut into three parts, each containing 85, 156, and 43 points; the first 85 and the last 43 points in each R-R cycle are assembled to form a 128-point segment; every four segments are grouped, and an n-level discrete wavelet transform (DWT) is performed to obtain the corresponding wavelet coefficients. Four of the computed wavelet coefficients are gathered as a wavelet vector.

The Euclidean distance between two wavelet vectors $S_{1}$ and $S_{2}$ is denoted as $d\left(S_{1},S_{2}\right)$; the intra- and inter-group distances can then be calculated using (6). Table 4 contains the outcomes of our QSMI design, with three one-dimensional algorithms for comparison. It is clear that the RBP, waveform, and wavelet transform algorithms perform well, with an approximately 75% success rate for human identification. But with an accuracy of 93%, the proposed QSMI algorithm using both ECG signal and blood oxygen level far outperforms the others.

### 5.2. Comparison with TTwo-d Dimensional MModels

Now, we compare the proposed QSMI scheme with two other commonly used two-dimensional algorithms: baseline and principal component analysis. 

(a) Baseline algorithm

The baseline algorithm is a simple and effective way to determine the similarity between two sparse matrices [35]. The sparse matrix is formed by the mapping rule and dimensionality reduction of the sample points. There are many standards of similarity. Here, we use the Euclidean distance; that is, we define the baseline to measure the similarity between two sparse matrices $SM_{1}$ and $SM_{2}$ as:(9)$$b=\underset{i}{\sum }\underset{j}{\sum }({\left(SM_{1}\right)}_{ij}-{\left(SM_{2}{)}_{ij}\right)}^{2}.$$

After the similarity is measured using (10), the correlation coefficient between two b’s, as well as the minimum and mean correlation coefficients $R_{min}^{i}$ and $R_{mean}^{i}$ for each individual *i*, can be computed. Furthermore, the FA, FR, and $\frac{\mathrm{FA}+\mathrm{FR}}{2}$ rates can be determined using (5).

(b) Principal component analysis (PCA) algorithm

PCA is a powerful tool for identifying similarities and differences in patterns of data. For high-dimension data, this method reduces the number of dimensions without much loss of information. PCA maps data in a high-dimensional space to a space of lower dimensions using a linear projection. This well-known dimension reduction method is commonly applied to investigate linear correlations across multiple time series [36].

For a matrix X consisting of sample data, a linear transformation converts X to Y through

(10)Y=PX

The linear mapping P allows Y to extract the principal components from X, and each row of P contains the eigenvectors of $C_{X}$, where

(11)$$C_{X}=\;\frac{1}{n}XX^{T}$$

In PCA assessment, we use the data collected from the previously mentioned wearable device, the difference being that the ECG signal is arranged prior to the blood oxygen level. The PCA algorithm extracts select features that are then used to train and test human identification. Experimental results confirm that the success rate for identification is much better with the five extracted features.

Figure 12 shows that with ECG signal and blood oxygen level combined, the proposed QSMI algorithm demonstrates a much lower $\frac{\mathrm{FA}+\mathrm{FR}}{2}$ value, which equates to a better accuracy rate than with the other two two-dimensional designs.

### 5.3. Comparison with MMulti-d Dimensional MModels

Here, we use PTB multi-lead ECG signal data. We read the multi-lead signal from the PTB data, fuse the multi-lead data signal, and then classify the individual. Similar to the experimental models in the previous two parts, the PTB database contains 249 individuals, and we divide the data into training sets and test sets. To evaluate the algorithm fairly and without loss of efficiency, we select 12 individual 12-lead signals to conduct our experiment. To analyze with other multi-dimensional feature algorithms, we compare the MDI algorithm with the PCA algorithm and the baseline-QSMI algorithm. The results are shown in Figure 13.

It can be seen from Figure 13 that the recognition rate of our MDI algorithm reaches 98.67%, which is about 3.4% higher than the recognition result with the baseline algorithm and up to 13% higher than with the PCA algorithm, thus proving that the final proposed MDI algorithm has considerable advantages.

### 5.4. Additional Discussion

This section briefly addresses two questions. Some studies conduct ECG analysis with common classifiers such as artificial neural networks and support vector machines [37]. Is such an approach suitable for this study? General classification algorithms are more commonly used in ECG diagnosis. Our research is more suitable for ECG identification, as it applies a one-to-one comparison of a similarity algorithm to identify a person. Because ECG is a one-dimensional data source, if it is used for many-to-many identification, high error rates may result.

Another question is whether a system combining ECG and photoplethysmography (PPG) data has practical value. Some studies have applied ECG and PPG to heart-rate analysis or diagnosis [38], and now smart watches are able to integrate these two sensors, so including diagnosis and identification devices in smart watches should be possible. The successful combination of light and electric sensors proven by our research can achieve a higher recognition rate, further increasing the application value of ECG and PPG.

## 6. Conclusions

This study presents an innovative and efficient approach to better verify human identification. Experiments confirm that the combination of ECG signal and blood oxygen level yields greater identification accuracy than using ECG signal alone. The identity verification success rate using the proposed QSMI algorithm is 93.14%, almost 20% higher than with other one-dimensional methods and 6% higher than with two commonly used two-dimensional algorithms. Practically speaking, all ECG and blood oxygen level data can be easily collected from a small wearable device and stored in the memory card of a mobile phone. If we can collect more data dimensions, we can increase the ECG recognition rate through our proposed MDI algorithm.

## Figures and Tables

**Figure 1 sensors-18-04138-f001:**
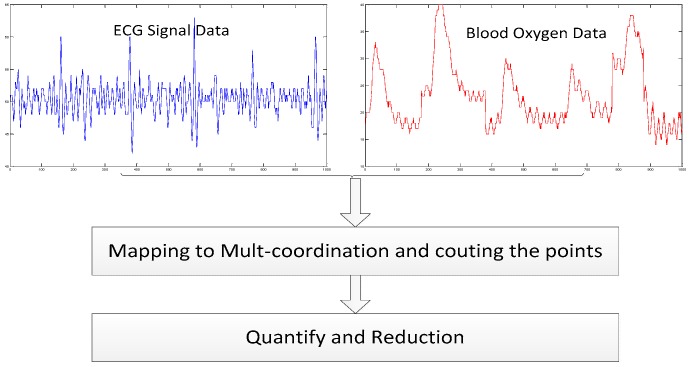
Flow of hybrid ECG and blood oxygen identification with QSMI algorithm.

**Figure 2 sensors-18-04138-f002:**
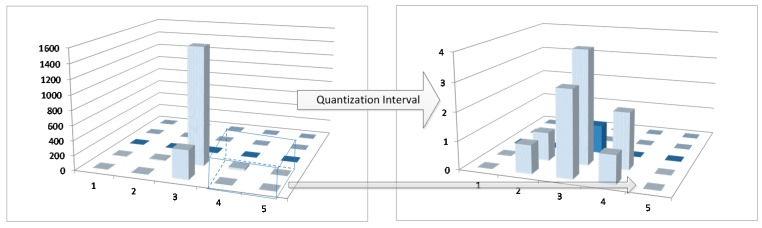
Procedure for 4 quantization levels with mask matrix $M^{*}$ of size 2×2.

**Figure 3 sensors-18-04138-f003:**
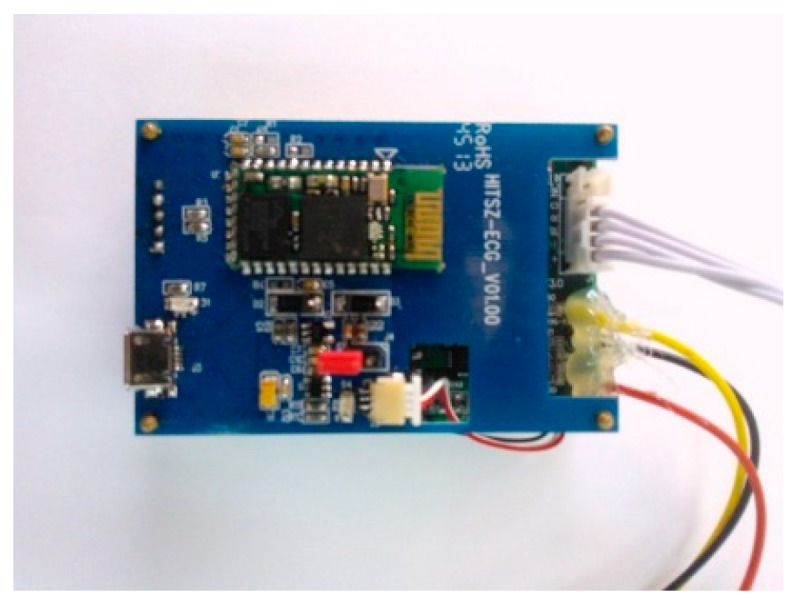
ECG and blood oxygen collection equipment.

**Figure 4 sensors-18-04138-f004:**
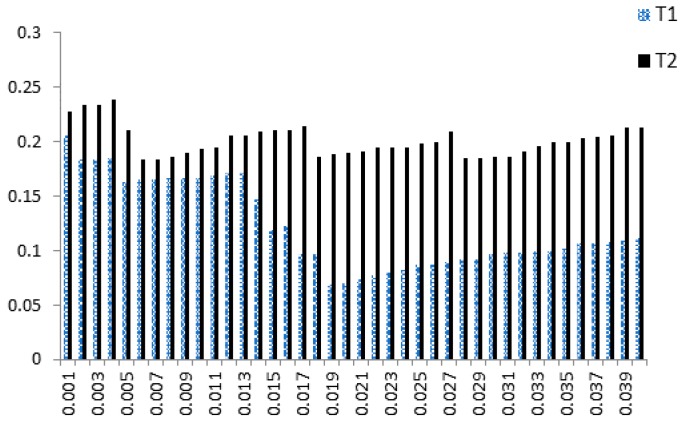
$\frac{\mathrm{FA}+\mathrm{FR}}{2}$ Rate of Each Template.

**Figure 5 sensors-18-04138-f005:**
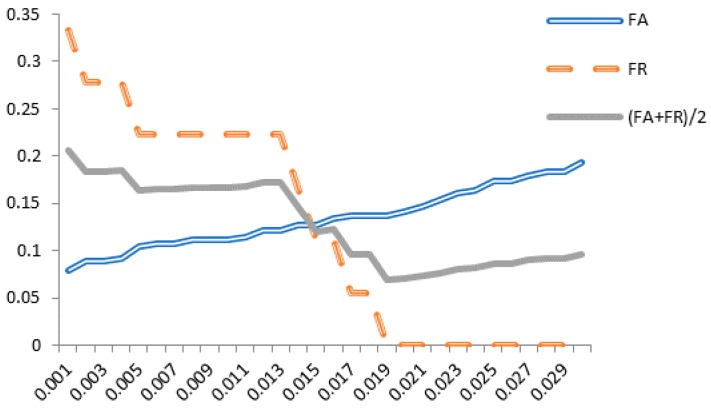
Detailed data for $T_{1}$ with δ.

**Figure 6 sensors-18-04138-f006:**
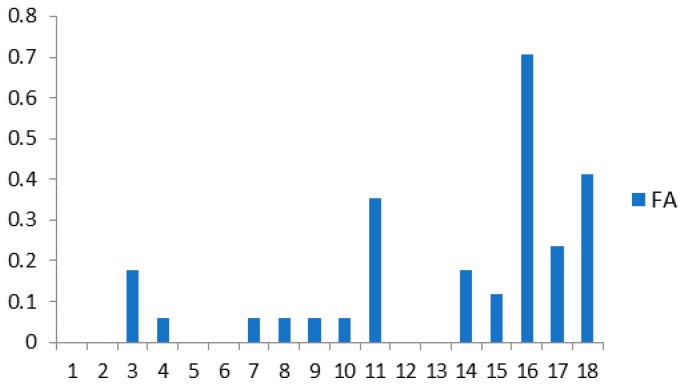
FA rate of each individual in *T*_1_.

**Figure 7 sensors-18-04138-f007:**
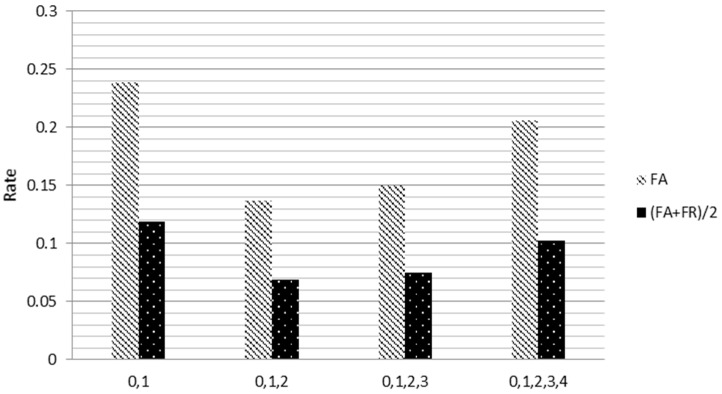
Comparison of different quantization intervals; three-level (0, 1, 2) quantization has the best performance with the lowest $\frac{\mathrm{FA}+\mathrm{FR}}{2}$ rate at 0.1193.

**Figure 8 sensors-18-04138-f008:**
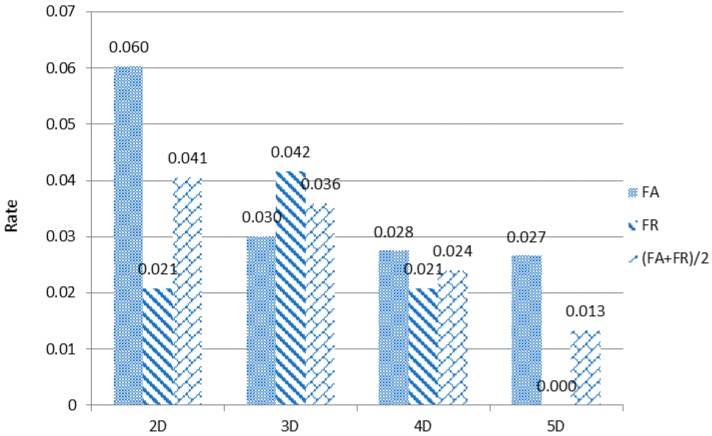
FA/FR rates from 2D to 5D.

**Figure 9 sensors-18-04138-f009:**
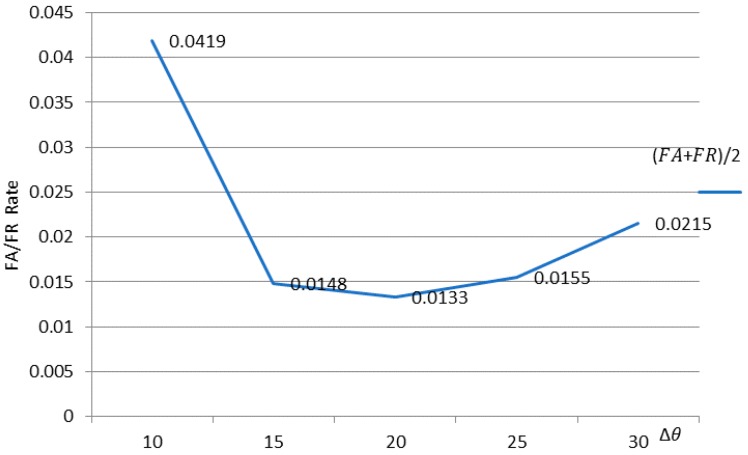
Change in ∆θ along the $\frac{\mathrm{FA}+\mathrm{FR}}{2}$ curve.

**Figure 10 sensors-18-04138-f010:**
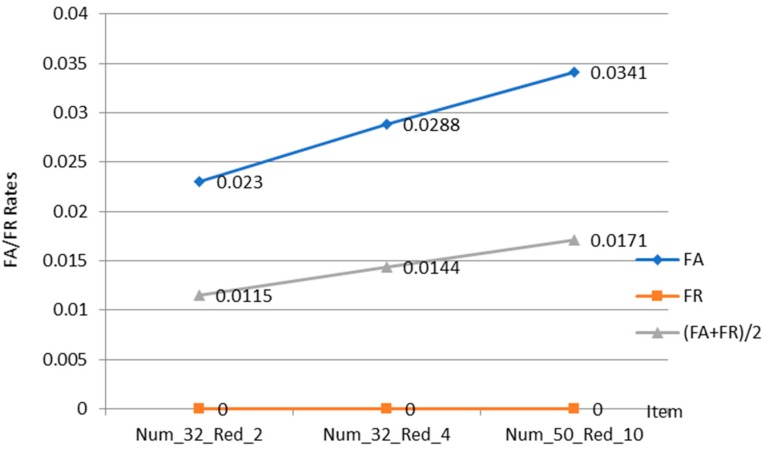
Comparison of DTW and Sparse Combination Parameters in 5D space.

**Figure 11 sensors-18-04138-f011:**
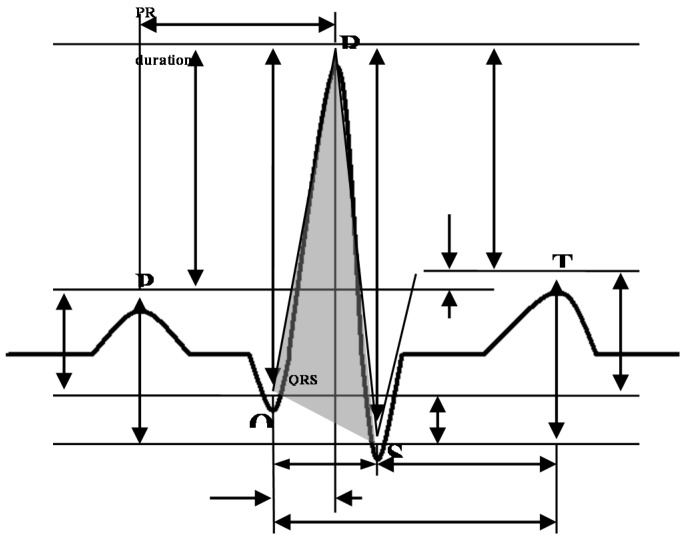
Waveform feature extraction for identification.

**Figure 12 sensors-18-04138-f012:**
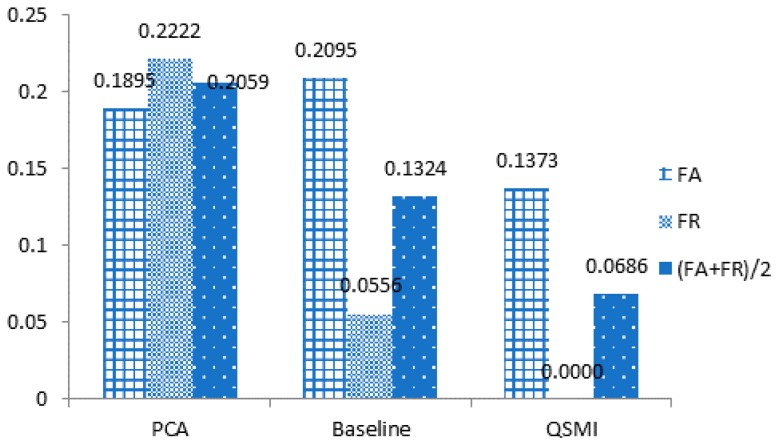
Comparison of ECG two-lead algorithms.

**Figure 13 sensors-18-04138-f013:**
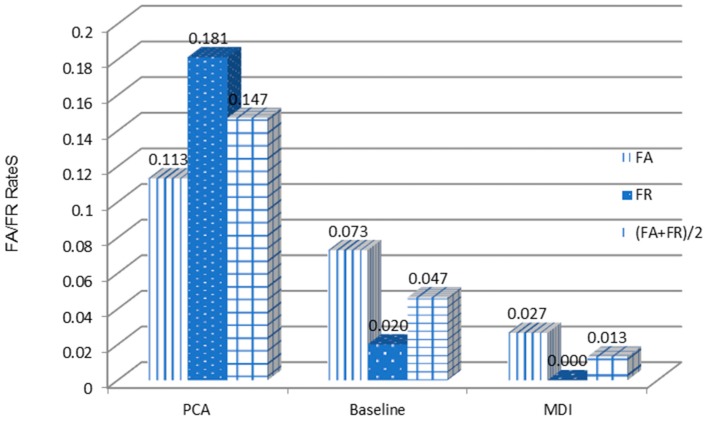
Multi-dimensional algorithm FA/FR comparison results.

**Table 1 sensors-18-04138-t001:** Reduce matrix.

	1	2	3	4	5
1	0	0	2	2	1
2	0	1	2	2	1
3	0	0	2	0	0
4	0	0	0	0	0
5	0	0	0	0	0

**Table 2 sensors-18-04138-t002:** Comparison of Memory Efficiency of DTW Combined Sparse Optimization Algorithm.

	Item	Num_32_Red_2	Num_32_Red_4	Num_50_Red_10	Baseline
Memory (bytes)					
Average (bytes)		2737	1307	675	25,000
Min		40	40	40	25,000
Max		13,920	7240	2640	25,000

**Table 3 sensors-18-04138-t003:** Time efficiency comparison of DTW with sparse optimization algorithm (time unit seconds).

	Item	Num_32_Red_2		Num_32_Red_4		Num_50_Red_10		Baseline	
Time(s)		Train	Test	Train	Test	Train	Test	Train	Test
1th		0.0966	1.5518	0.0223	0.3732	0.0060	0.0972	0.0044	0.0083
2th		0.0953	1.5780	0.0229	0.3643	0.0059	0.0984	0.0013	0.0082
3th		0.0950	1.5680	0.0226	0.3655	0.0059	0.0994	0.0014	0.0084
Average		0.0956	1.5659	0.0226	0.3677	0.0059	0.0983	0.0024	0.0083

**Table 4 sensors-18-04138-t004:** Comparison of FA, FR, and $\frac{\mathrm{FA}+\mathrm{FR}}{2}$ rates.

Item	RBP	Waveform	Wavelet	QSMI
FA	0.4880	0.2418	0.5196	0.1373
FR	0	0.2222	0	0
(FA + FR)/2	0.2440	0.2320	0.2598	0.0686
Accuary	75.60%	76.8%	74.02%	93.14%

Next, we compare our scheme with two other typical two-dimensional algorithms.

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
