# Peer review of "Electrocardiograph Identification Using Hybrid Quantization Sparse Matrix and Multi-Dimensional Approaches"

_sensors, 2018, doi:10.3390/s18124138_

Reviewer 1 Report

I would like to have some information on  the device shown in Fig 3 (I am an hardware guy).

Is it commercial? Can you provide a reference with performances or 

characteristics ? or a datasheet ? 

If your group designed it, has it been described in other publications ? 

Can you provide a reference ?

In fig 7  if FR=0 (FA+FR)/2 does not provide any more information with respect to FA (in fact it is simply FA/2). I think you can add an explanation to the caption of fig 7

Your conclusion is interesting, in effect is more interesting that you state...

I suggest you to add that your experiments demonstrated that blood oxygen provides an independent (from ECG) source of information for individual identification. In fact, table 4 shows that using only ECG data you have a 25% error rate (1/4). With two independent source of information (ECG + blood oxygen) you obtain about 1/4*1/4 error rate (near to 6%).

Author Response

Reviewer 1:

I would like to have some information on the device shown in Fig 3 (I am an hardware guy). Is it commercial? Can you provide a reference with performances or characteristics ? or a datasheet ?

Answer:

This is not a commercial device, it is our experimental board, there is a label HITSZ-ECG_V01.00 on it. HITSZ is a short name of our university. The following is a enlarge picture for the label.

If your group designed it, has it been described in other publications? Can you provide a reference?

Answer:

Yeah, this board is designed by our group; it has not been described in other publication. This board is mainly using TI AFE4900 chip, The AFE4900 device is an analog front-end (AFE) for synchronized electrocardiogram (ECG),photoplethysmogram (PPG) signal acquisition. The device can also be used for optical bio-sensing applications, such as heart-rate monitoring (HRM) and saturation of peripheral capillary oxygen (SpO2).

The following is the detailed specification for TI AFE4900 Specification:
Synchronized PPG, ECG Signal Acquisition at Data Rates Up to 1 kHz
• ECG Signal Chain:
– Standalone ECG Acquisition Up to 4 kHz
– Input Bias: 1-Lead ECG With RLD Bias
– Programmable INA Gain: 2 to 12
– Input Noise (1 Hz to 150 Hz): 2.5 μVrms at1 kHz, 1.25 μVrms at 4-kHz Data Rate
– AC, DC Lead-Off Detect: 12.5-nA to 100-nA
• PPG Receiver:
– Supports Three Time-Multiplexed PD Inputs
– 24-Bit Representation of Current From PD
– DC Offset Subtraction DAC (Up to ±126-μA) atTIA Input for Each LED, Ambient
– Digital Ambient Subtraction at ADC Output
– Noise Filtering With Programmable Bandwidth
– Transimpedance Gain: 10 kΩto 2 MΩ
– Dynamic Range Up to 100 dB
– Receiver Operates in PPG-Only mode atApproximately 1-μA/Hz Sampling Rate
– Power-Down Mode: Approximately 0 μA
• PPG Transmitter:
– Four LEDs in Common Anode Configuration
– 8-Bit LED Current Up to 200 mA
– Mode to Fire Two LEDs in Parallel
– Programmable LED On-Time
– Simultaneous Support of Three LEDs forSpO2, or Multiwavelength HRM
– Average Current of 30 μA Adequate for aTypical Heart-Rate Monitoring Scenario:
– 20-mA Setting, 60-μs Pulse Duration,25-Hz Sampling Rate
• Clocking Using an External or Internal Clock
• FIFO With 128-Sample Depth for ECG and PPG
• I2C, SPI™ Interfaces: Selectable by Pin
• Operating Temperature Range: –20℃ to +70℃ 
• 2.6-mm × 2.1-mm, 0.4-mm Pitch DSBGA Package
• Supplies:
– Rx: 1.8 V to 1.9 V (LDO Bypass),
2.0 V to 3.6 V (LDO Enabled)
– Tx: 3 V to 5.25 V
– IO: 1.7 V to Rx_SUP

In fig 7 if FR=0 (FA + FR)/2 does not provide any more information with respect to FA (in fact it is simply FA/2). I think you can add an explanation to the caption of fig 7

Answer:

Yeah, there is why we show the FA and (FA+FR)/2 both in figure 7. We have added the explanation on title of Figure 7.

Your conclusion is interesting, in effect is more interesting that you state...

Answer: Thanks for reviewer’s agreement.

I suggest you to add that your experiments demonstrated that blood oxygen provides an independent (from ECG) source of information for individual identification. In fact, table 4 shows that using only ECG data you have a 25% error rate (1/4). With two independent source of information (ECG + blood oxygen) you obtain about 1/4*1/4 error rate (near to 6%).

Answer: According our experiment, the performance is poor with blood oxygen only for individual identification. Because the waveform of blood oxygen is relatively smooth and not prominent. According to our preliminary experiment, the single blood oxygen sensor has around 60% recognition rate, so there is no further need for this testing. However, according to our experiments, the combination of ECG and blood oxygen has good practical performance, since ECG has different characteristics with blood oxygen, it will greatly improve the performance from ECG alone.

Reviewer 2 Report

The authors described an innovative biometric data collection algorithm based on usage of ECG signal and blood oxygen level properly arranged in such matrix representation.

The method seems very promising according to the reported comparison results. Moreover, the proposed approach is careful detailed in the paper.

Just some advices to improve the understandability/robustness of the proposed pipeline:

1. Add more details about the used DTW approach. The authors referred a reference to describe the used algorithm but it is not well clear how DTW

reduces the computational complexity by increasing the recognition rate. Please add more details.

2. The authors introduce DTW and MDI in the paragraph "Advanced Model" without any preliminary description but referring to next paragraphs for details. This way to describe the proposed pipeline is not very well as the reader has to switch up/down in the paper in order to understand the role of each algorithm. Please, changes the information flow of the paper accordingly;

3. The author described the used ECG and blood oxygen collection equipment. Anyway the ECG collection requires a lot of preprocessing and signal stabilization not recalled in the paper. Please describe better this device and the method for obtaining a compliant ECG signals comparing the used approach with some prior art. I suggest a comparison with the following papers as reports a good review on that field:

Sansone M., Fusco R.Pepino A,. Sansone C. “Electrocardiogram pattern recognition and analysis based on artificial neural networks and support vector machines: a review.” Journal of healthcare engineering 4 4 (2013): 465-504.

Rundo, F.; Conoci, S.; Ortis, A.; Battiato, S. An Advanced Bio-Inspired PhotoPlethysmoGraphy (PPG) and ECG Pattern Recognition System for Medical Assessment. Sensors 201818, 405. 

4. About used thresholds (QSMI algorithm), please add a sensitivity analysis. Moreover, since the authors reported experiments in 18 individuals divided part for training and part for testing, they have to describe the ratio between training/validation set.

Author Response

Reviewer 2:

The authors described an innovative biometric data collection algorithm based on usage of ECG signal and blood oxygen level properly arranged in such matrix representation. The method seems very promising according to the reported comparison results. Moreover, the proposed approach is careful detailed in the paper.

Just some advices to improve the understandability/robustness of the proposed pipeline:

1. Add more details about the used DTW approach. The authors referred a reference to describe the used algorithm but it is not well clear how DTW reduces the computational complexity by increasing the recognition rate. Please add more details.

Answer:

Thanks, we have added more details on DTW approach in 3.2 Advanced Model. Compared dimension.

2. The authors introduce DTW and MDI in the paragraph "Advanced Model" without any preliminary description but referring to next paragraphs for details. This way to describe the proposed pipeline is not very well as the reader has to switch up/down in the paper in order to understand the role of each algorithm. Please, changes the information flow of the paper accordingly;

Answer: Thanks, In section 3.2, we have modified it, we have add introduction to DTW and MDI. Also the following paper is added for the understanding of multi-dimensional DTW.

39.   M. Shokoohi-Yekta, B. Hu, H. Jin, J. Wang and E Keogh, Generalizing dtw to the multi-dimensional case requires an adaptive approach, Data Mining & Knowledge Discovery, 31(1), 1-31, 2017.

3. The author described the used ECG and blood oxygen collection equipment. Anyway the ECG collection requires a lot of preprocessing and signal stabilization not recalled in the paper. Please describe better this device and the method for obtaining a compliant ECG signals comparing the used approach with some prior art. I suggest a comparison with the following papers as reports a good review on that field:

Sansone M., Fusco R., Pepino A,. Sansone C. “Electrocardiogram pattern recognition and analysis based on artificial neural networks and support vector machines: a review.” Journal of healthcare engineering 4 4 (2013): 465-504.

Rundo, F.; Conoci, S.; Ortis, A.; Battiato, S. An Advanced Bio-Inspired PhotoPlethysmoGraphy (PPG) and ECG Pattern Recognition System for Medical Assessment. Sensors 2018, 18, 405.

Answer:

Thanks, we have added a descriptive comparison for the above two articles in section 5.4 Other Discussions.

4. About used thresholds (QSMI algorithm), please add a sensitivity analysis. Moreover, since the authors reported experiments in 18 individuals divided part for training and part for testing, they have to describe the ratio between training/validation set.

Answer: We are sorry; maybe the previous description is not clear enough. What we do is one-to-one comparison. Therefore, 18 people should be enough for training and testing. For training and testing sets, ECG of each person is collected multiple times for our experiments. At the same time, in order to get a reasonable threshold, we are doing threshold calculation in training. In addition, the other 17 people’s data will be treated as unknown ECG for training and testing. Thus, our experiment should have a similar to sensitivity analysis.

Reviewer 3 Report

In this paper a method for biometric identification based on ECG and blood oxygen level is presented.

Although I am not a native English, when I begun to read the paper I found myself quite confused because I was not able to understand certain phrases. I got to understand more or less section 1 and 2, but I was not capable of understand methodology. I only read over next sections because it is obvious that this paper can not be accepted in the present form, it needs a complete revision by a native or a people with a good level of English.

Some additional recommendations to improve the paper:

- Ref [1] must be moved to the end of this paragraph

- DTW initials must be explained

- William H [22] must be changed to jackson [22]

- Too many references to aspects not related to this paper. For instance [23-25] and [26,27]. Please select only some of them.

- Methodology section must be improved, not only for the English, but also notation. I found myself very confused  when. For instance, is equation (2) correct?

- line 147. Mathworks documentation?

- line 159 PTB database?

- equation 4  Is correct? I cannot understand what else mean.

Author Response

Reviewer 3:

In this paper a method for biometric identification based on ECG and blood oxygen level is presented. Although I am not a native English, when I begun to read the paper I found myself quite confused because I was not able to understand certain phrases. I got to understand more or less section 1 and 2, but I was not capable of understand methodology. I only read over next sections because it is obvious that this paper can not be accepted in the present form, it needs a complete revision by a native or a people with a good level of English.

Answer: Thanks for reviewer’s comment; we had edited the paper by an English native.

Some additional recommendations to improve the paper:

- Ref [1] must be moved to the end of this paragraph

Answer:

Thanks, we have modified it.

- DTW initials must be explained

Answer:

Thanks, We have added Dynamic Time Warping (DTW) in section,

- William H [22] must be changed to jackson [22]

Answer:

Thanks, we have modified it.

- Too many references to aspects not related to this paper. For instance [23-25] and [26,27]. Please select only some of them.

Answer: Thanks, we have selected the suitable references.

- Methodology section must be improved, not only for the English, but also notation. I found myself very confused when. For instance, is equation (2) correct?

Answer: Thanks, we have modified this problem.

- line 147. Mathworks documentation?

Answer: Thanks, we delete this extra sentence.

- line 159 PTB database?

Answer: We have updated the weblink, PTB database (www.physionet.org/physiobank/database/ptbdb/) is a popular database for Diagnostic ECG.

- equation 4 Is correct? I cannot understand what else mean.

Answer: Thanks, we have added more explanation to this equation.